# Behavior of Steel–Coconut Shell Concrete–Steel Composite Beam without and with Shear Studs under Flexural Load

**DOI:** 10.3390/ma13112444

**Published:** 2020-05-27

**Authors:** Lakshmi Thangasamy, Gunasekaran Kandasamy

**Affiliations:** Department of Civil Engineering, Faculty of Engineering and Technology, SRM Institute of Science and Technology, Chennai 603203, India; lakshmitny@gmail.com

**Keywords:** double-skin steel plates, coconut shell concrete, composite beam, shear studs, flexural behavior

## Abstract

In this study, we investigated using coconut shell concrete (CSC) in double-skin steel plate sandwich beams, i.e., steel–concrete–steel (SCS) under flexure. Two cases—without and with shear studs to interconnect the bottom tension and top compression plates—were considered. Conventional concrete (CC) was used for comparison purposes. The effect of quarry dust (QD) in place of river sand (RS) was considered. Therefore, four mixes named as CC, conventional concrete produced using QD (CCQ), CSC and coconut shell concrete produced using QD (CSCQ) were used. Three different steel plate thicknesses were considered (4 mm, 6 mm and 8 mm). In total, twelve SCS specimens were tested to evaluate the flexural performance under two-point static loads. Study parameters include: partial and fully composite, ultimate moment and failures, deflection characteristics, ductility property, cracking behavior and strains in both tension and compression plates. It was found that the moment carrying capacity of the SCS sandwich beams increased when the thickness of the steel plate increased. Our results provided evidence that using QD in place of RS augmented the strength of beams. Theoretical deflections were underestimated the experimental deflection, except in one case. The SCS beams showed good ductility behavior. The SCS beams exhibited crack widths at yielding well below guideline values.

## 1. Introduction

There is a new form of sandwich construction, consisting of two thin steel plates with a concrete infill built into the construction of submerged-tube tunnels by Narayanan et al. [1]. Much research has also been carried out on using steel–concrete–steel (SCS) forms in recent years [2,3,4,5]. Some significant works carried out in this SCS sandwich structures, as well as different concrete materials, are mentioned here. JB Yan et al. (2020) studied SCS sandwich composite structures using novel enhanced C-channels and ultra-high performance concrete (UHPC) [6]. JB Yan et al. (2020) also developed sandwich composite beams with novel J-hooks using UHPC, i.e., steel–UHPC–steel sandwich composite beams with J-hooks [7].

Generally, SCS consists of a layer of unreinforced concrete core, sandwiched between two relatively thin steel plates (top compression plate and bottom tension plate), that are connected to the concrete infill through welded shear stud connectors (Figure 1).

Plates are aligned in the plane of bending so they are in compression and tension, respectively when subjected to flexure. No further shear reinforcement is necessary in the SCS form construction. Shear studs are provided for this purpose. This SCS type of construction is very economical. It combines the advantages of both steel and concrete material properties. It is very suitable for the variety of structures like building cores, tube tunnels, gravity sea walls, floating breakwater, caissons, nuclear structures, liquid containment and blast resistant structures [8].

Traditional reinforced concrete elements may be failed due to crushing of concrete. Yielding of steel may be under flexure, shear and torsion or due to bond failure. Likewise, in SCS elements also the collapse may be triggered because of any one mode like, yielding of steel in tension, yielding or buckling of steel in compression, crushing of concrete in compression, tension steel slip, shear failure of concrete or failure of shear studs [8]. In these, yielding of steel in tension or compression and the crushing of concrete in compression are governed by the choice of the appropriate material and their physical properties. Some of the significant failure modes of SCS element are illustrated in Figure 2.

Many research works were carried out on this SCS form type of construction and few glimpses of them are discussed here. Zhenyu Huang and Richard Liew (2016) investigated the structural behavior of SCS sandwich walls filled with ultra-lightweight cementitious composite materials. They reported that the SCS sandwich walls with J-hook connectors exhibit comparable behavior in compressive resistance and post-peak unloading behavior as ones with overlapped headed studs [9]. Sohel et al. (2012) used novel shear connectors such as J-hook and cable shear connectors and their performance to achieve composite strength of the SCS sandwich structures. They stated that the use of these connectors together with ultra-lightweight cement composite core reduces the overall weight of SCS sandwich system and makes it suitable for the construction of marine and offshore structures. They carried out static tests on SCS sandwich beams with J-hook, cable shear connectors and headed studs and reported the ultimate strengths and also about their respective failure modes [10]. Liew and Sohel (2009) examined a new concept in composite structures consisting of a lightweight concrete (LWC) core sandwiched between top and bottom steel plates connected by J-hook. They conducted a push out test and found that the J-hook connectors have the capacity superior to the conventional headed stud connector in achieving composite action [11].

### 1.1. Significance of SCS and CSC

Studies were carried out by many researchers on this SCS form of construction and used with different concrete materials. However, use of coconut shell concrete (CSC) as a core material on this SCS form construction and their results are very limited. Literatures also stated that the yielding of steel in tension or compression and the crushing of concrete in compression in this SCS form of construction are governed by the choice of the appropriate material and their physical properties. The CSC is established by using CS as coarse aggregate in the production of concrete and also the study on CSC incorporated as a core concrete in SCS form of construction and their properties are very limited. Therefore, in this study use of CSC as core material in SCS form was taken significantly to study about its behavior in SCS form of construction. Since CSC mostly falls into the category of LWC, some significant details of LWC are presented here. Since the main material used in this study work is CSC, the general properties, durability properties, behavior of CSC are also presented in this section. In addition, since quarry dust (QD) was also used as an alternate material for river sand (RS), some details about QD also presented. Overall, the most significance of this study is to use CSC as core material in the SCS form of construction. Hence, this study focused on the behavior of steel–CSC–steel composite beam without and with shear studs under flexural load. With and without shear studs were taken because shear studs also play a significant vital role in contributing the performance of SCS beams.

### 1.2. Lightweight Concrete

Lightweight concrete (LWC) is characteristically geared up by using natural or man-made lightweight aggregates (LWA) or by entraining air into a concrete mixture. Several LWA used for LWC productions are pumice, perlite, expanded clay or vermiculite, coal slag, sintered fly ash, rice husk, straw, sawdust, cork granules, wheat husk, oil palm shell, palm kernel shell and coconut shell (CS) [12,13,14]. the compressive strength of LWC ranges from 7 N/mm^2^ for light partition to about 40 N/mm^2^ for the primary structural components. Natural LWA is normally obtained from the volcanic rocks. Artificially contrived LWAs are obtained from diverse natural materials such as expanded clay, expanded shale, foamed slag, blast furnace slag, pulverized fuel ash and perlite to enable LWC and to achieve compressive strength up to 100 N/mm^2^ [15]. In countries where copious agricultural and industrial wastes are discharged can also be used as probable or alternate material in the construction industry.

### 1.3. Coconut Shell Concrete Properties in General

In recent times, crushed CS is used as coarse aggregate in the production of concrete. It is one of the sustainable replacement materials for the conventional crushed stone aggregate in concrete production to some extent. Several works were carried out and reported on CSC in the past. Some of the properties of CSC are highlighted here: it has better workability, density in the range of structural LWC, flexural and split tensile strength is comparable with conventional concrete (CC), impact resistance of CSC is high compared to CC and bond strength of CSC is also comparable with CC and other LWC [16]. CS is to be treated for its water absorption before it is used in concrete. CSC performed well for its compressive strength in intermittent curing conditions compared to full water immersion. CS aggregate does not deteriorate due to its age of CSC. There is no bond failure in CSC at later ages and its long term performance is well in quality [16]. Plastic shrinkage area on CSC slabs decreased when the percentage of CS increased. Deflections of CSC slab increased when percentage of CS increased. Compared to CC, CSC gives warning before it fails [17].

### 1.4. CSC Durability Properties

Pore structures presented in CS helped to observe water and enhanced the hydration of CSC in the early age of specimens. Proper curing is necessary for CSC for its better durability performance. Volume of permeable pore voids are more in CSC compared to CC. Sorptivity of CSC is comparable with the concrete produced using similar to CS such as expanded shale, sintered pulverized fuel ash and oil palm shell. There is moderate chloride-ion penetrability in CSC. Chloride concentration is quite high at the surface level of CSC specimens and consistently decreases with increased depth from surface to interior. Resistance at elevated temperature and the residual strength of CSC is comparable with other LWC. Overall, durability studies on CSC indicated that CS is a sustainable alternate material for conventional coarse aggregate in concrete [16].

### 1.5. CSC Beam Behavior

Coconut shell concrete beam behavior under flexure is comparable with the other types of LWC beams. Failures of CSC beams are not because of bond failure between the aggregate and the reinforcement. Deflections and crack widths of CSC beams under service loads are within the acceptable limits as per the standards and showed good ductility behavior. Under flexural loading CSC is able to achieve its full strain capacity [16]. CSC beams with and without shear reinforcements failed in flexure and diagonal shear modes, respectively. CSC beams exhibit higher ductility ratios compared to CC beams with and without shear reinforcements, respectively [16]. Resistances of CSC beams are more compared to CC beams under torsion as well. A minimum of 1% volumetric torsional reinforcement is needed in CSC beams to avoid sudden failure. For the same amount of volumetric torsional reinforcements, CSC beams exhibited more ductility than CC beams [18].

### 1.6. Quarry Dust

The concept of replacing the natural fine aggregate by QD could improve the utilization of generated QD. Thus, reducing the land filled area required and conserving the scarcely available natural RS for sustainability. The suitability of QD as a sand replaced material shows that the mechanical properties and also elastic modulus are improved. Replacement for sand by the QD induced higher compressive strength [19].

### 1.7. Need for the Study

Compared to conventional stone aggregate, CS has less stiffness and strength. In addition, naturally CS has higher porosity and permeability which results in higher water absorption and less durable performance. If concrete has low modulus of elasticity, it will lead to inferior mechanical performance [20]. Flexural behavior of reinforced concrete is controlled generally by the provision of steel reinforcement and therefore structural performance is not significantly affected by the mechanical and material properties used in concrete. Though many studies have taken SCS elements using different types of concrete with different parameters [2,3,4,5], its combination with CSC is not reported so far. Therefore, this study is taken to study the behavior of double-skin steel plate CSC composite beams under flexure.

### 1.8. Research Significance

At first, double-skin steel filled with concrete without shear connectors was studied to understand the basics. Subsequently, in SCS elements studies many types of shear connectors such as angle–steel bar–angle, angle–T channel, angle–steel hoop–angle, angle–C channel–angle, U connector–steel bar–U connector, angle–I beam–angle, angle–angle, root connector, U connector–steel cable–U connector and J-hook connectors [10] were used in general to develop effective bond between the steel plates and the concrete core. However, this study is only the starting picture in the study of combining SCS and CSC; simple steel bars without heads were used as shear studs. Results were compared with CC used SCS beams without and with shear studs.

## 2. Materials and Properties

Ordinary Portland cement (OPC) conforming to IS: 12269-2013 [21], river sand (RS) as fine aggregate conforming to grading zone III as per IS: 383-2016 [22] was used in both CSC and CC production. QD fine particles passing 4.75 mm were used as it is which conformed to grading zone IV as per IS 383: 2016 was used as a replaced material for RS in both CSC and CC. Crushed CS and crushed granite stone (CGS) of maximum size 12.5 mm was used in the production of CSC and CC. CS was used in saturated surface dry (SSD) before it was added in the CSC mixes. Table 1 shows the properties of aggregates used and Table 2 illustrates the mix proportion adopted. To study the effect of QD in both CSC and CC, equivalent volume of RS was replaced by QD in concrete appropriately. Then the equal volume of QD weighed to arrive the mix proportion. QD used CSC is designated as CSCQ and QD used CC is designated as CCQ. Table 3 illustrates the properties of four concrete used.

The modulus of elasticity is one of the most important parameters in structural concrete as it is required for assessing deflections and cracking of a structure. In general, LWC has a lower modulus of elasticity compared to its normal weight counterpart. It has been found that the modulus of elasticity for LWC is lower by approximately 25–50% than normal weight concrete for the same strength. In the case of CSC there was a softening stress–strain trend curve compared to CC. This is happening because of less stiffness of CS compared to conventional coarse aggregate. One particular concern for CSC is the low value of modulus of elasticity, caused by the shock absorbent nature of the CS aggregates. CS has high resistance against impact and crushing. However, it was found that the modulus of elasticity of CSC is comparable to other LWC of similar compressive strength. During CSC cylinder tested for finding the modulus of elasticity, it was also observed that for CSC, the crack propagation length at failure was about half of the height of the specimen. For the CC cracks were observed throughout the entire length of the specimen. The failure mode for CSC suggested that only local failure occurred, and it again demonstrated the good energy absorbing capacity of the concrete [16]. Since this modulus of elasticity of CSC and its comparison is not the main focus and hence further discussion on this is restricted in this study.

Mild steel (MS) plates were used as top compression and bottom tension plate to develop the sandwich beam. Length and width of steel plate used was 2400 mm and 150 mm, respectively. Overall, the depth of SCS beam was taken as 230 mm for the convenience of testing of beam in the loading frame facility available in the institute premises. A typical beam with top and bottom MS plate maintained with gauge rod without shear studs and with shear studs are shown in Figure 3 and Figure 4, respectively. Three different thicknesses of MS plates were considered to study the effect of thickness of MS plates. The thickness of steel plate was selected as 4 mm, 6 mm and 8 mm.

Since this is a pilot and an initial stage study, it was attempted to use conventional steel bars without head for two points. One is to use the conventional steel bar as shear stud connectors to make a simple process to get shear connectors like normal bar cutting for the ready use at the site. Second one is to get the results data for its performance compared to shear studs with head in future. Therefore, the geometry of the shear connectors used in this study was only conventional steel bars (diameter and length) cut into required length and used and not made any complicated geometry like normal shear studs or connectors. Shear connectors in the form of studs without a head were welded. of Eight millimeter-shank-diameter and 165-mm-high Fe 415-grade studs were used as the shear studs. Stud-spacing of 150 mm and cover of 25 mm were maintained as shown in Figure 4.

## 3. Analytical Prediction

Flexural behavior of composite beam using a theoretical elastic and plastic approach, shear capacity of shear stud connectors and deflection were predicted analytically to compare the experimental results. Analytical prediction methodology and the respective equations used are discussed in this section.

### 3.1. Flexural Behavior of Composite Beam—A Theoretical Elastic Approach

Like conventional method of design assumption, here also the concrete tensile strength contribution is neglected while calculating flexural resistance of double-skinned sandwich composite section. Furthermore, ignored the relative stiffness of thin steel plates about their own axes under flexure. The stresses developed on steel plates and concrete during the testing are assumed to be elastic and linearly distributed in the section of beam. To prevent the local buckling of the compression plate, shear studs were closely spaced to provide lateral restraint. Figure 5a illustrates the section of SCS beam. Figure 5b shows the equivalent steel section. Figure 5c shows both compressive and tensile forces distributed in SCS beam section and Figure 5d shows the idealized stress distribution in SCS beam section, respectively. With the assumptions mentioned above, from equivalent section) Figure 5b the theoretical position of neutral axis *z*, using the basic conventional design principle equation suggested by Liew et al. (2009) [11] and it is given in Equation (1):(1)$$z=-m\left(t_{c}+t_{t}\right)+{\left[m^{2}{\left(t_{c}+t_{t}\right)}^{2}-m\left(t_{c}{}^{2}-2t_{t}h_{c}-t_{t}{}^{2}\right)\right]}^{1/2}$$
where, *m* = modular ratio [Ratio of the modulus of elasticity between steel (E_s_) and concrete (E_c_)]. Figure 5c,d illustrates distribution of force in section and stress distribution in ideal condition. It is assumed that the normal stress distribution is linear throughout the depth of concrete core h. The flexural resistance of a SCS sandwich composite section can be determined by taking the moments of the line of action of the concrete compressive force.

The moment of resistance at yielding of the sandwich section was calculated using the equation suggested by Liew et al. (2009) [11] and it is given in Equation (2), in which it is assumed that the first yield occurs at the tension plate (i.e., σ*_t_* = σ*_y_*) and also the beam is fully composite.
(2)$$M_{y}={bt}_{c}\left(\frac{z}{3}+\frac{t_{c}}{2}\right)\;\sigma_{y}\frac{z+t_{c}/2}{h_{c}-z+t_{t}/2}+\sigma_{y}{bt}_{t}\left(h_{c}-\frac{z}{3}+\frac{t_{t}}{2}\right)$$

To develop fully composite action, enough shear connectors should be provided to resist the maximum longitudinal force generated in the steel face plate. However, the number of shear connectors depends upon the individual capacity of the shear studs used in the core concrete. If, N_cs (max)_ is considered as maximum longitudinal force generated in the steel face plate, then it can be calculated by using the basic principle of force calculation, $\sigma_{y}$*b*$t_{t}$ (stress × area). This N_cs (max)_ should be resisted by the capacity of shear studs available between the points between zero and maximum moment for full composite.

If N_t (max)_ is considered as maximum tensile force in the bottom plate, then it can be calculated as N_t(max)_ = *n_s_*·*P_Rd_* for full composite action and N_t(max)_ = *n_p_*·*P_Rd_* for partial composite action. Moreover, the maximum tensile force in the bottom plate can be calculated as σ*_t_* = $\frac{n_{p\;P_{RD}}}{bt_{t}}$.

where, *n_s_* = number of shear studs between the points zero and maximum moment for full composite action.

*n_p_* = number of shear studs between the points zero and maximum moment for partial composite action.

*P_Rd_* = Shear resistance of shear studs within the concrete core.

The moment resistance for a partially composite sandwich beam can be calculated using equation suggested by Liew et al. (2009) [11] and it is given in Equation (3);
(3)$$M_{p}=n_{p}P_{RD}\;\left[\left(\frac{t_{c}}{t_{t}}\right)\left(\frac{z+t_{c}/2}{h_{c}-z+t_{t}/2}\right)\left(\frac{z}{3}+\frac{t_{c}}{2}\right)+\left(h_{c}-\frac{z}{3}+\frac{t_{t}}{2}\right)\right]$$

### 3.2. Flexural Behavior of Composite Beam—A Theoretical Plastic Approach

It is assumed that the concrete below the neutral axis to be cracked and also no buckling in the compression steel plate in a plastic approach. If the *t_c_* = *t_t_* = *t*, then the plastic moment of resistance for fully composite action can be calculated using equation suggested by Liew et al. (2009) [11] and it is given in Equation (4);
(4)$$M_{ult}=\sigma_{y}bt\left(h_{c}+t\right)$$

Moreover, the plastic moment of resistance of the partially composite beam section can be determined using equation suggested by Liew et al. (2009) [11] and it is given in Equation (5);
(5)$$M_{pl}=n_{p}P_{RD}\left(h_{c}+t\right)$$

### 3.3. Shear Capacity of Shear Stud Connector

Eurocode 4 [23] gives the equations to predict the strength of shear studs (used with normal concrete and LWC and those are given in Equations (6) and (7), respectively.

For shear studs used with normal concrete Equation (6)
(6)$$P_{RD}=0.8\;\sigma_{u}\;\left(\frac{\pi }{4}d^{2}\right)/\;\gamma_{v}$$
and for shear studs used with LWC Equation (7):(7)PRD=0.29αd2fckEcmγv
where, d = diameter of the stud shank; σ*_u_* = ultimate tensile strength of the stud (≤500 MPa); *f_ck_* = characteristic cylinder strength of concrete; *E_cm_* = secant modulus of concrete; α = 0:2(*h_s_*/*d* + 1) for 3 ≤ *h_s_*/*d* ≤ 4 or α = 1:0 for *h_s_*/*d* ≥ 4; *h_s_* = overall height of the stud. The recommended partial safety factor γ*_v_* for the connector is 1.25. For the studs used in this study and with its properties, the values for different mixes used are given below:$P_{RD}$ = 15.60 kN for CC and CCQ mixes$P_{RD}$ = 6.45 kN for CSC and 6.82 kN CSCQ mixesNumber of studs required, *n_s_* = Equation $\sigma_{y}$*b*$t_{t}$/$P_{RD}$.

### 3.4. Deflection

The theoretical calculation of flexural deflection of SCS sandwich beam subjected to two point-load is discussed in this section. Transfer of shear deformation often dominates if the span to thickness ratio is small. The bond strength between the steel plate and core concrete influences significantly the flexural stiffness of the sandwich beam. There is an appropriate method suggested by Roberts et al. [24] a reduction factor to allow for slip by reducing the effective stiffness of the steel plates. If, *k_t_* and *k_c_* are the stiffness reduction factors for the tension steel plates and compression steel plates, then those are given in Equations (8) and (9), respectively.
(8)$$k_{t}=\frac{n_{a}K}{n_{a}K+2bt_{t}E_{s}/L}$$
and
(9)$$k_{c}=\frac{n_{a}K}{n_{a}K+2bt_{c}E_{s}/L}$$
where, *n_a_* = Number of shear connectors provided between maximum moment and zero moment and *K* = Stiffness of the shear connectors determined from the push-out test (*K* = 27,820 N/mm for CC, *K* = 28,140 N/mm for CCQ, *K* = 24,620 N/mm for CSC and *K* = 24,990 N/mm for CSCQ mixes). (Note that the procedure and the results of push-out test are not given in detail, only values are given since it is not a main scope of this manuscript.) Then, the beam deflection (Δ) due to the two point load of the span can be calculated using the basic deflection principle equation given in Equation (10).
(10)$$\Delta =\frac{Wa}{24\;D}\left(3L^{2}-4a^{2}\right)$$
where, *D* = (EI) equivalent is the flexural stiffness of the composite section. An equivalent moment of inertia (for a sandwich beam considering a cracked section can be calculated from the basic principle applied in the Equation (11) and hence ‘D’ using the Equation (12).
(11)$$I_{eq}=\frac{bK_{c}t_{c}{}^{3}}{12}+\left(bk_{c}t_{c}\right){\left(z+\frac{t_{c}}{2}\right)}^{2}+\frac{\left(\frac{b}{m}\right)z^{3}}{3}+\frac{bK_{t}t_{t}{}^{3}}{12}+\left(bk_{t}t_{t}\right){\left(h_{c}-z+\frac{t_{t}}{2}\right)}^{2}$$
then,
(12)$$D=E_{s}\left\{\frac{bK_{c}t_{c}{}^{3}}{12}+\left(bk_{c}t_{c}\right){\left(z+\frac{t_{c}}{2}\right)}^{2}+\frac{\left(\frac{b}{m}\right)z^{3}}{3}+\frac{bK_{t}t_{t}{}^{3}}{12}+\left(bk_{t}t_{t}\right){\left(h_{c}-z+\frac{t_{t}}{2}\right)}^{2}\right\}$$

## 4. Experimental Investigation

Test specimen’s details, number of specimens, concreting, cast and test methodology are given in this section.

### 4.1. Test Specimens

Twelve SCS sandwich beam specimens tested under two point loading. Gauge rods (Tata steel, Chennai, India) used at random to maintain the overall depth of beam as 230 mm. The thickness of the face plates 4 mm, 6 mm and 8 mm were used in this study. Shear studs using 8-mm-diameter Fe 415 steel bars of length equal to 165 mm were welded on both top (compression) and bottom (tension) plates at 150 mm center-to-center throughout the length of beam. The cover of 25 mm was maintained on all the sides of the beam.

Both CC and CSC beams without and with shear studs were cast. Curing was done on all the beam specimens covered with gunny bags for 28 days as followed in traditional site curing. During testing of the beams were placed in such a way that the tension plate and compression plates were kept at top and bottom, respectively.

### 4.2. SCS Beam Testing

For the plate, 4-mm, 6-mm and 8-mm CC, CCQ, CSC and CSCQ, mixed used beams were tested under a two-point load. Totally 12 beams were tested. All the beams were white cemented over concrete surfaces except the plates for marking the positions of supports, load points, dial gauges (TMC Associates, Chennai, India) and linear voltage displacement transducer (LVDT) (Aimil, Chennai, India). Beam to be tested was lifted and positioned in the loading frame. Level tube was used for the orientation of the beam position before testing. Simple supports were provided at either side with the bearing of 100 mm hence the effective span of the beam fixed as 2200 mm. Ten-millimeter length electrical strain gauge TML (Tokyo Measuring Instruments Lab, Tokyo, Japan) (resistance of strain gauge was 120 Ω) was used to measure the strains developed at the center of tension plate and compression plate and at the center of concrete depth of beam. All the strain gauges were connected to the 10 channel data logger. An LVDT and dial gauge were used at quarter span of beam specimen on either side at bottom and at the center of beam bottom to measure deflection. With these set up, a spreader beam of required size was placed over the beam to distribute the load at the middle third points of the beam. Proving ring was placed over the hydraulic jack capacity 50 T placed over the spreader beam. Now the beam specimen was made ready for testing with the entire set up. A schematic diagram of beam test setup is shown in Figure 6. A typical experimental set up is shown in Figure 7.

## 5. Discussion on Test Results

In this section, test results on behavior of SCS beam element without and with shear studs, ultimate moment and failures, deflection characteristics, ductility properties of SCS beams, cracking behavior of SCS beams and strains on tension and compression plates are discussed.

### 5.1. Behavior of SCS Beam Element without Shear Studs

Double skin SCS beam element without shear studs was lifted and placed in the loading frame. Since shear studs were not provided gauge rods were not removed until testing was started to maintain the integrity between the steel plates and the core concrete. Just before the start of the test, gauge rods were removed in a gentle manner as shown in Figure 8. It was noticed and observed that the debonding of both tension and compression steel plates were started while grinding to remove the gauge rod itself and it is shown in Figure 9.

However, with the debonded plate flexural test was continued on the beam. In continuation of the load applied, debonding of plates extended and the slip of the tension plate has taken place near the support as shown in Figure 10. Once the slip of the tension plate occurs, the concrete element fails suddenly in a brittle manner at the flexural zone as shown in Figure 11. In all cases of CC, CCQ, CSC and CSCQ beam elements the ultimate load was observed to be 9.81 kN and all the specimens were failed at the flexure zone. The bottom tension plate slip was measured in range of 100–110 mm. Since all the specimens slipped and failed suddenly as expected, for safety consideration of the instrument and the person, the central deflection and strain measurements on steel plates and concrete were not taken in this typical study.

### 5.2. Behavior of SCS Elements with Shear Studs

Shear stud connectors employed in SCS sandwich beams resist any slip or separation between steel plate and concrete, there is no further shear reinforcement needed. Figure 12, Figure 13 and Figure 14 show a SCS element before the start of test, during testing and after testing, respectively. All the beams were tested for their flexural strength and the other parameters include strains on compression plate, tension plate and also on concrete.

### 5.3. Ultimate Moment and Failures

Both the elastic and plastic theories were used to predict the moment capacity of the composite beams and summarized in Table 4. Predicted theoretical and experimental ultimate moments of all twelve SCS beams are consolidated and presented in Table 4. Since the predicted moment capacity of SCS beams obtained using the plastic theory approach is always higher than that predicted using elastic theory, only plastic moments and experimental moments are considered for the calculation of capacity ratios and presented in Table 4. Experimental shear capacity of the studs was used to predict the ultimate moment capacity of the beams. The predicted ultimate moment is generally conservative in all the SCS beams tested.

The failure was started with the initial separation of plate due to slight local buckling followed by yielding of the plates. The first crack was observed on the specimen in the flexure zone followed by the appearance of other cracks propagated upon further increase in load. Flexural failure was indicated in a gentle crushing of compression concrete and yielding of tension steel were observed in all specimens. The predicted ultimate moment is generally conservative, i.e., the predicted maximum moment of the beam is lower than the experimental results. The ultimate moment carrying capacity of CCQ is higher than the other beams. This is due to the strength of the concrete core. The strength of the LWC core was lower than that of the CC. The test result shows that the moment carrying capacity of SCS sandwich beams increased when the thickness of the steel plate increased.

The experimental ultimate moment capacity of 4-mm, 6-mm and 8-mm plate used beams were 2%, 6%, 107% and 115%; 13%, 38%, 125% and 122%; 36%, 64%, 133% and 138%; higher for CC, CCQ, CSC and CSCQ beams than predicted moments, respectively. It can be seen that the moment capacity of CSC and CSCQ mixes are more than 100% higher compared to theoretical moments predicted even though the strength and stiffness of CSC and CSCQ mixes are lesser than CC and CCQ mixes. The reason for this more capacity ratio may be due to low modulus of elasticity of CSC and CSCQ compared to CC and CSCQ mixes. The same trends were found in case of CSC beams studied the behavior under flexure, shear and torsion as well [16]. One more reason may be that the established equations used to predict both elastic and plastic theory approaches are based on the conventional concrete material properties. Because of using those equations for this CSC for comparison purpose also the reason for difference of more capacity ratios because the material properties also play a vital role while in establishing the design parameters. Further research is needed to elaborate in this context. However, this should be considered and investigated further in future works to conform. However, here it can be stated that use of SCS technique has advantages compared to conventional reinforced concrete technique. Since in this SCS technique steel plates are placed both top and bottom covering throughout length of the element uniformly instead of covering the steel reinforcement by cover concrete in case of conventional reinforced concrete, the steel plates taking responsibility much for the distribution of moment applied compared to concrete core may be the reason for higher moment capacity. This shows that both the thickness of plate and the concrete strength plays vital roles for the moment capacity of SCS beams and the same are reflected in the equation used for moment calculation. It can be noted that except for one case where CC mix used with 4 mm plates all other cases fell under partially composite action. Compared to CC and CSC mixes used beams, CCQ and CSCQ beams offered more moments of resistance. This shows that the use of QD in place of RS augmented the beam strength and hence QD can be considered as a sustainable alternate material for RS.

### 5.4. Deflection Characteristics

Moment–deflection characteristics were found in similar fashions in all beams. Figure 15, Figure 16, Figure 17 and Figure 18 shows the moment–deflection curves for SCS beams tested with 4-mm, 6-mm and 8-mm-thickness plates used with CC, CCQ, CSC and CSCQ mixes, respectively.

It was found that the moment–deflection curve of all the beams are approximately steep and linear up to 65% to 70% of its ultimate capacity until formation of crack occurs. Once the cracks formed, this curve became nonlinear. During this period cracks widths are increases. Therefore, the central deflection observed approximately at the two-third of the ultimate moment was taken as the basis and compared the theoretical deflection. The same thing was recommended in the literature as well [11]. The reason for this consideration is stated in the literature as “the flexural stiffness reduces in case of SCS beams when the cracks are formed in concrete core and also the nonlinearity of moment–deflection characteristics is dependent upon the cracks extension and yielding of steel plates” [11]. The predicted deflections and its comparison with experimental deflections are presented in Table 5.

In post-cracking stage, the moment–deflection curve indicated that there is more deflection with the increase of load compared to the pre-cracking stage. From the pattern of moment–deflection characteristics, it can be observed that the SCS beams used with CSC and CSCQ beams show the similar behavior of SCS beams used with CC and CCQ mixes. It can be noted that except for the cases where 4 mm plates are used all other cases were underestimated the experimental deflection compared to theoretical deflection. Almost all the beams were experienced yielding of bottom tension plates as failure mode. Further, it should be noted that the experimental investigations were done on only short-term loadings and did not allow for shrinkage and creep of concrete.

### 5.5. Ductility Property of SCS Beams

Deformation ductility (μ) is the ratio between ultimate deflection (Δ*_u_*) and the deflection at yielding of steel plates (Δ*_y_*). That is, µ =$\;\frac{\Delta_{u}}{\Delta_{y}}$. If the ductility ratio is high, then it is considered that the particular structural member is able to withstand large deformations and give warning before it fails. If the deformation ductility ratio is in the range between 3 and 5, then it can be considered adequate for structural elements subjected to more deformations due to sudden forces caused by dynamic nature [16]. Deformation ductility values of the tested SCS beams of this study are presented in Table 6.

It can be seen from Table 6 that the ductility ratio was more than 4 in all the cases of SCS beams used indicated that they are relatively more ductile. It can also be noted that the irrespective concrete mixes (strength) all SCS beams have ductility ratio more than 4 which also indicated that the plate used contributed more for ductility behavior compared to concrete strength. It is also an added advantage in case of SCS construction technology compared to convention reinforced concrete techniques where concrete strength and aggregate strength also play a significant role in ductility property [16].

### 5.6. Cracking Behavior of SCS Beams

Once the cracks formed, crack width at every increment of load was measured at the junction of bottom tension plate and the concrete. Cracks formed were marked on the beam. It was observed that the first crack appeared always on all SCS specimens in almost the central flexural zone (i.e.) close to the mid span. Vertical crack formed almost at the central pure flexural zone of the beams in the bottom of the concrete core Very fine hair cracks appeared horizontally at the junction of tension plate and concrete core (interfacial zone) in all the tested beams and it was happened nearly 50% of the ultimate capacity of the beams. Those horizontal interfacial cracks were extended towards support ends. Once the load applied increased the number of cracks formed in the concrete core. This happened in all the beams nearly at 65% to 70% of the ultimate capacity. No beams experienced any significant shear cracks formation indicated that the studs used in this study are able to resist shear developed in the beams and offered enough force transfer mechanism between the concrete core and steel plates. Crack values of all the SCS beams tested are given in Table 7.

There should not be any excessive cracking to state that the structural element performs satisfactorily. If the cracks are wide then it will allow ingress of any harmful substances into that element. Excessive cracks may also look unsightly and cause concern to the users. Therefore, various standards recommended that the maximum crack widths should lie within the range between 0.1 to 0.4 mm on different exposure conditions [16]. The standards—both IS 456:2000 [25] and BS 8110 [26] guidelines—state that the surface crack width should be limited to 0.3 mm. However, this guideline value is pertaining to conventional reinforced concrete structures. However, this crack width recommendation is suggested for the durability aspect and hence the same was considered for this SCS beam study also. Considering in light of this, all the SCS beams tested exhibited the crack widths at yielding were well below that guideline value stipulated in IS 456:2000 [25] and BS 8110 [26] for durability aspect. The average spacing of cracks of 4-mm, 6-mm and 8-mm plate used beams were between 218–230, 137–206, 149–198 and 143–173 mm for the mixes CC, CCQ, CSC and CSCQ, respectively.

### 5.7. Tension and Compression Plate’s Strains

Strains at bottom tension and top compression plates were measured for every increment of load applied. The strain distributions of bottom tension plate and top compression plate for all the mixes CC, CCQ, CSC and CSCQ are shown in Figure 19, Figure 20, Figure 21 and Figure 22, respectively. Results of tension and compression steel plates’ strain of tested beams at yielding is presented in Table 8.

From Figure 23, it can be seen that, as the thickness of plates increases strain also increases in CC and CCQ mixes. In general, for a particular concrete strength if the cross sectional area of steel increases strain may be less for a particular level of load applied. However, in this study the load applied until the beam’s ultimate capacity. Therefore, the strains are more when the thickness of plate increases as the moment capacity also increases. It shows that the use of CC in SCS beam elements behave typically. At the same time, this trend was reversed in case of CSC and CSCQ mixes used SCS beams. The possible reason for this trend is that the moment carrying capacity of CSC and CSCQ beams are less compared to CC and CCQ mixes. However, in case of compression plates strain (Figure 24) neither the CC and CCQ mixes nor the CSC and CSCQ mixes trends do not happen as in the case of bottom tension plate strains. The possible reason for this there may be an influence of strain due concrete uncertainty (though the concrete is strong in compression, it is generally heterogeneous in nature). However, further research is needed to find the proper reason to get better conclusion on this.

## 6. Conclusions

This study proposes a concept of using CSC into the double-skin steel plates, i.e., SCS composite sandwich beam elements. The SCS composite beams were tested in both without and with shear studs. In case of double-skin SCS beam element without shear studs, initially, debonding of both tension and compression steel plates happened. Debonding of plates extended and the slip of the tension plate has taken place near the support. Once the slip of the tension plate occurs, the concrete core fails suddenly in a brittle manner at the flexural zone. This phenomenon happened for all the SCS beam elements irrespective of the different concrete mixes used. In case of double-skin SCS beam element with shear studs, the predicted ultimate moment is generally conservative in all the SCS beams tested. The failure started with the initial separation of plate due to slight local buckling followed by yielding of the plates. The ultimate moment carrying capacity of CCQ mix is higher than the other beams due to the strength of the concrete core. The test result shows that the moment carrying capacity of SCS sandwich beams increased when the thickness of the steel plate increased. This study shows that the use of QD in place of RS augmented the strength of beams. Hence QD can be considered as a sustainable alternate material for RS. The deflection characteristics of SCS beams produced with CSC and CSCQ beams show the similar behavior of SCS beams used with CC and CCQ mixes. The theoretical deflections were underestimated the experimental deflection except for the cases of 4 mm plates used. All the SCS beams showed good ductility behavior. No beams experienced any significant shear cracks formation indicated that the studs used in this study are able to resist shear developed in the beams. All the SCS beams exhibited the crack widths at yielding were well below that guideline value. If the thickness of plate increases, the strain also increases in both CC and CCQ mixes, showing that the uses of CC in SCS beam elements behave typically. At the same time, this trend was reversed in CSC and CSCQ mixes using SCS beams. However, all parameters considered on SCS beams produced with CSC and CSCQ mixes were in parallel with CC and CCQ mixes used SCS beams. Hence, CS and QD can be considered as the sustainable alternate material for conventional coarse aggregate and RS, respectively in the production of concrete.

## Figures and Tables

**Figure 1 materials-13-02444-f001:**
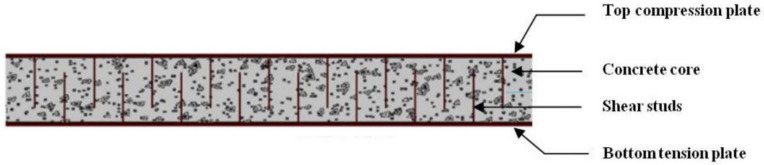
Illustration of typical steel-concrete-steel sandwich beam.

**Figure 2 materials-13-02444-f002:**
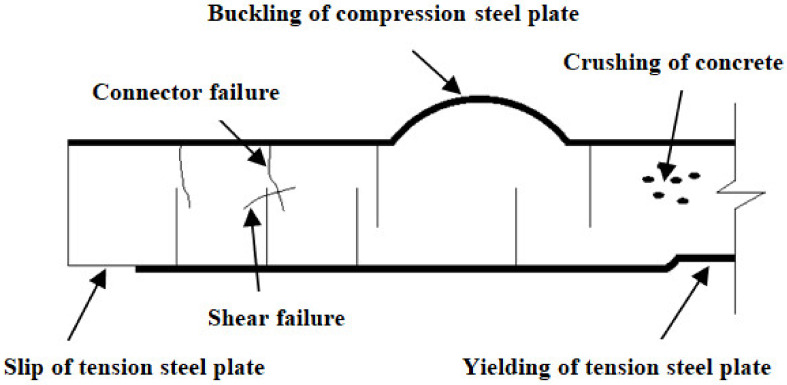
Typical failure modes of SCS beam element.

**Figure 3 materials-13-02444-f003:**
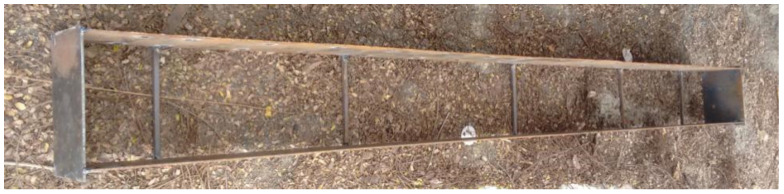
Beam specimen without shear studs.

**Figure 4 materials-13-02444-f004:**
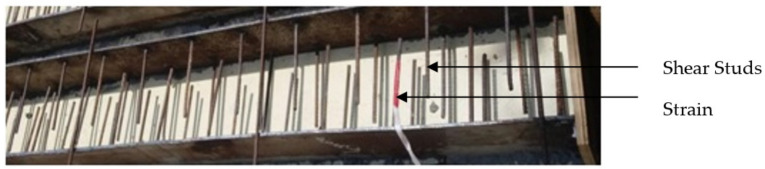
Beam specimen with shear studs.

**Figure 5 materials-13-02444-f005:**
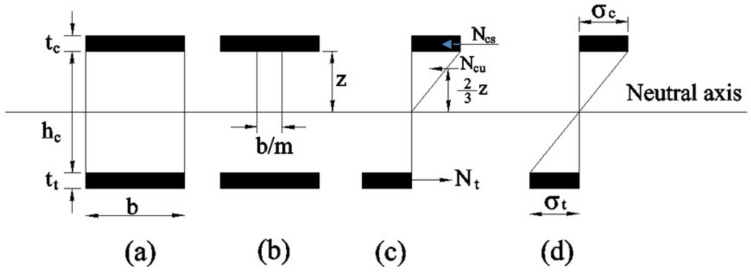
Sectional details for flexural resistance of SCS beam. (**a**) Section of (SCS) beam; (**b**) Equivalent section; (**c**) Compressive and tensile forces distribution (**d**) Idealized stress distribution

**Figure 6 materials-13-02444-f006:**
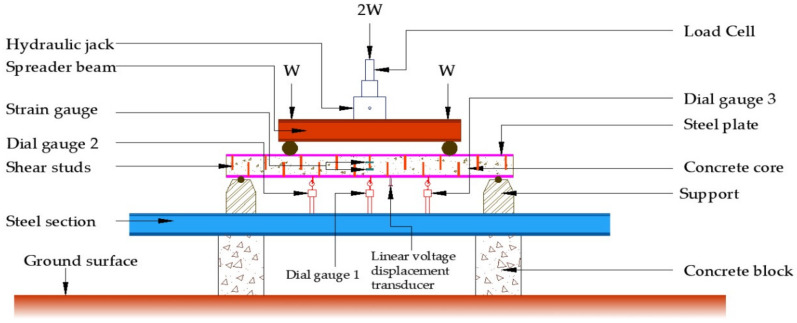
Schematic diagram of testing arrangement.

**Figure 7 materials-13-02444-f007:**
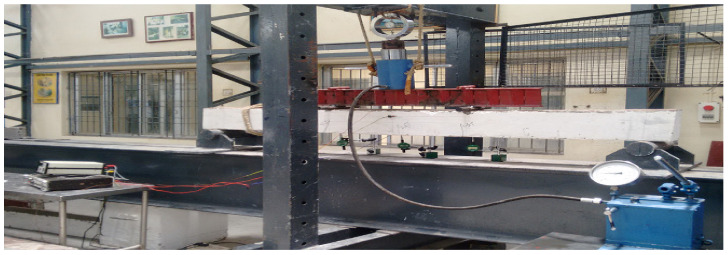
Typical experimental testing arrangement.

**Figure 8 materials-13-02444-f008:**
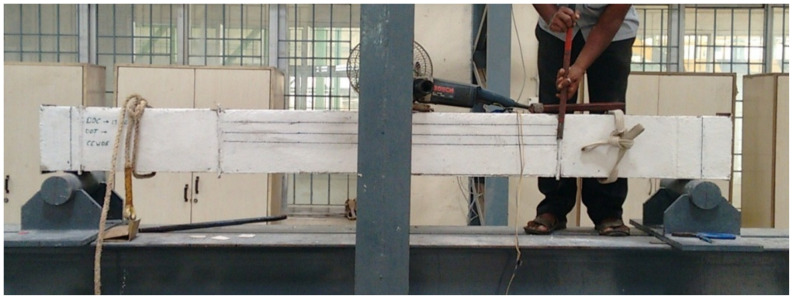
Removal of gauge rod in a SCS element without shear studs.

**Figure 9 materials-13-02444-f009:**
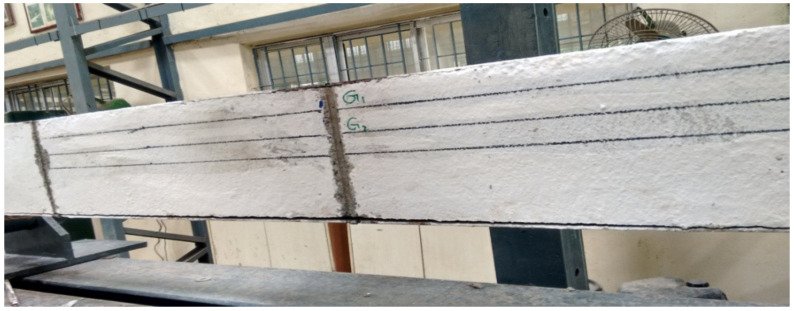
Compression and tension plate debonded.

**Figure 10 materials-13-02444-f010:**
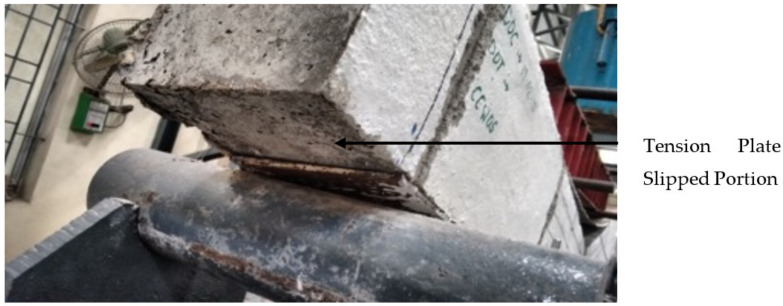
Slip of tension plate near the support.

**Figure 11 materials-13-02444-f011:**
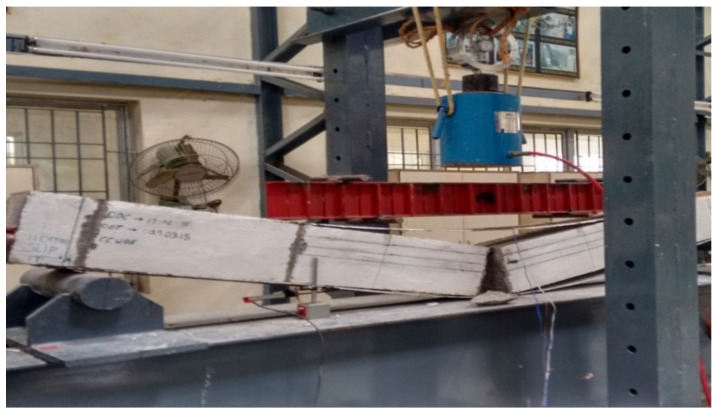
Specimen failed at flexural zone.

**Figure 12 materials-13-02444-f012:**
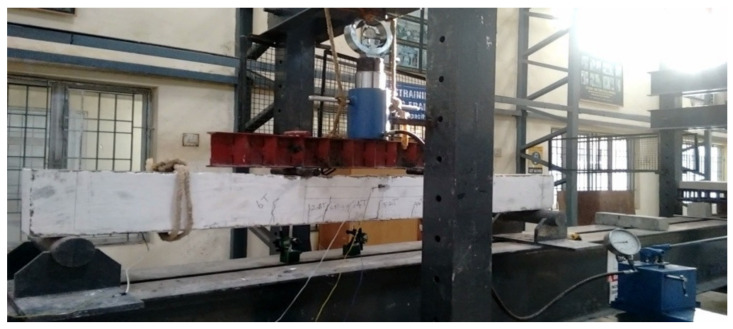
SCS beam with shear studs under flexural test (before testing).

**Figure 13 materials-13-02444-f013:**
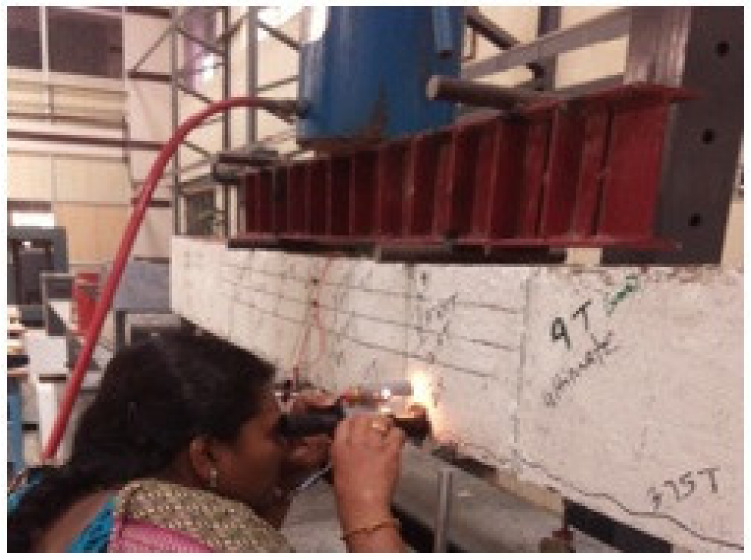
SCS beam with shear studs under flexural test (during testing).

**Figure 14 materials-13-02444-f014:**
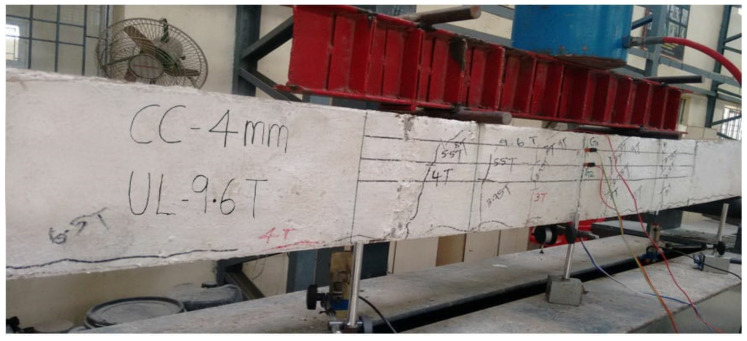
SCS beam with shear studs under flexural test (after testing).

**Figure 15 materials-13-02444-f015:**
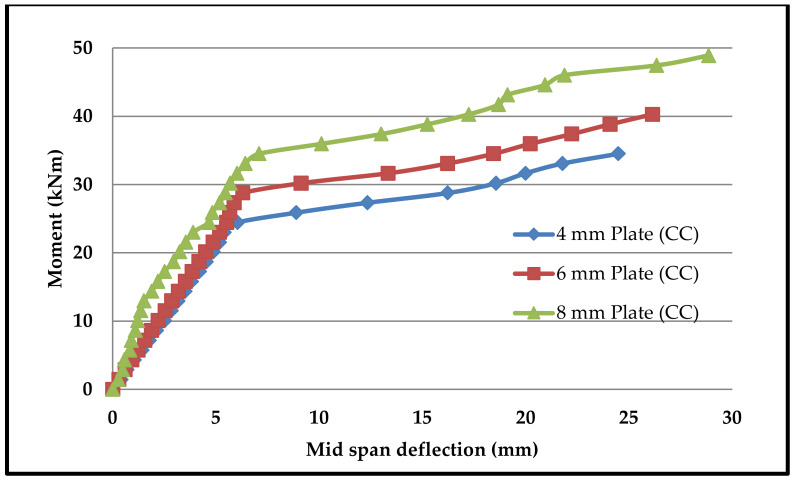
Moment–deflection curve for SCS beams used with conventional concrete (CC) mix.

**Figure 16 materials-13-02444-f016:**
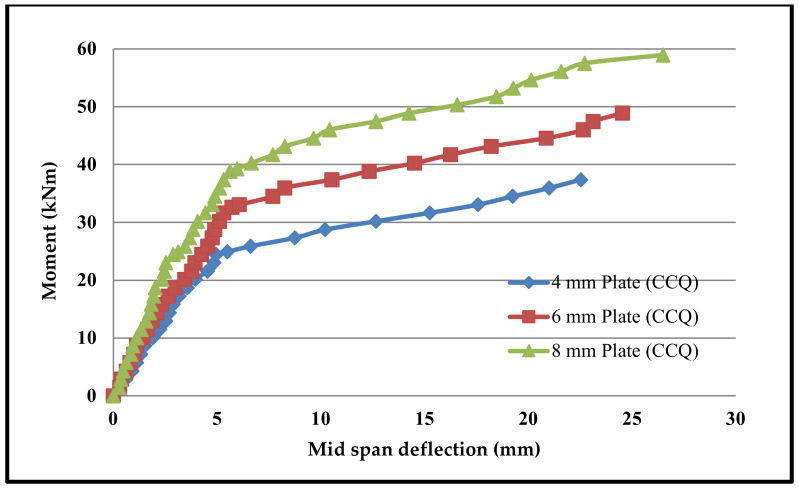
Moment–deflection curve for SCS beams used with CCQ mix.

**Figure 17 materials-13-02444-f017:**
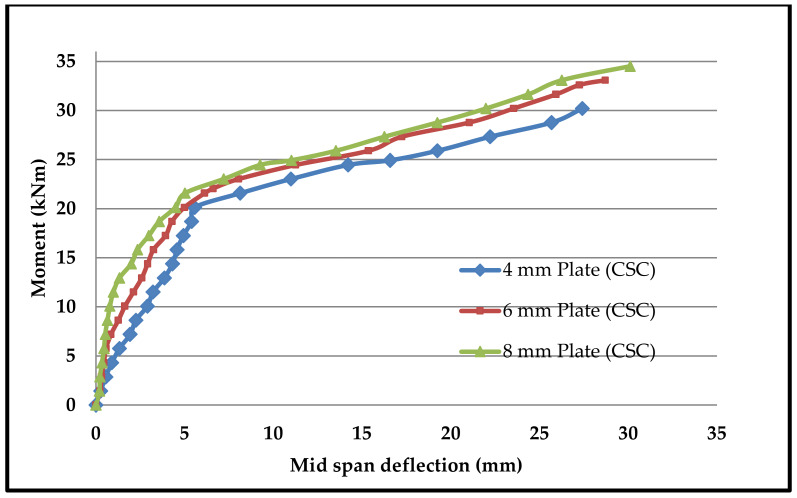
Moment–deflection curve for SCS beams used with CSC mix.

**Figure 18 materials-13-02444-f018:**
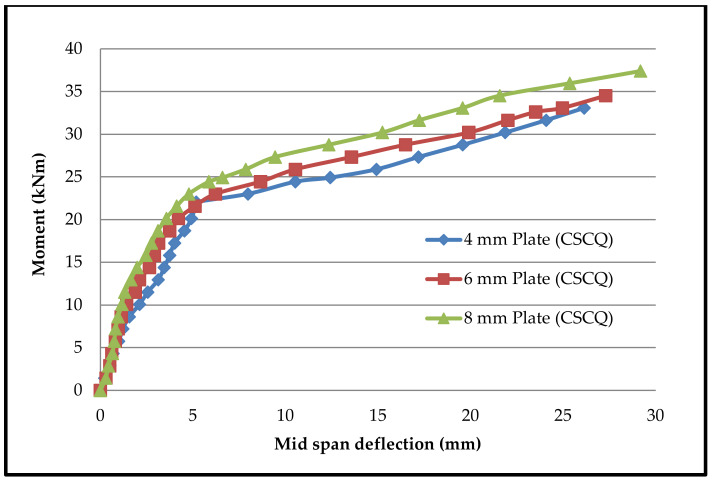
Moment–deflection curve for SCS beams used with CSCQ mix.

**Figure 19 materials-13-02444-f019:**
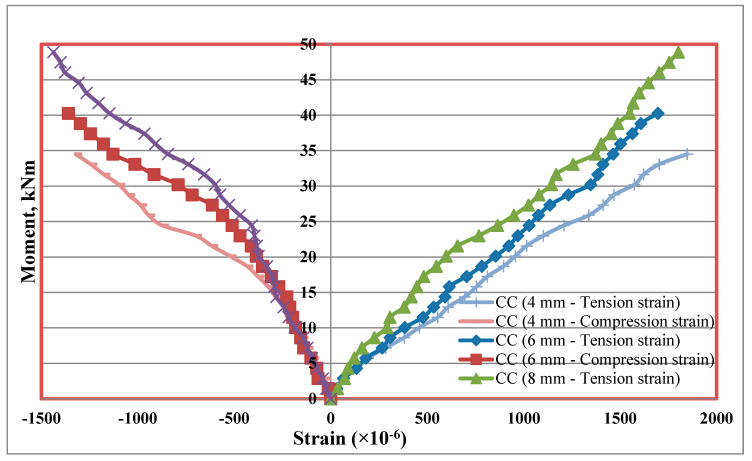
Strain distribution of SCS beams used with CC mix.

**Figure 20 materials-13-02444-f020:**
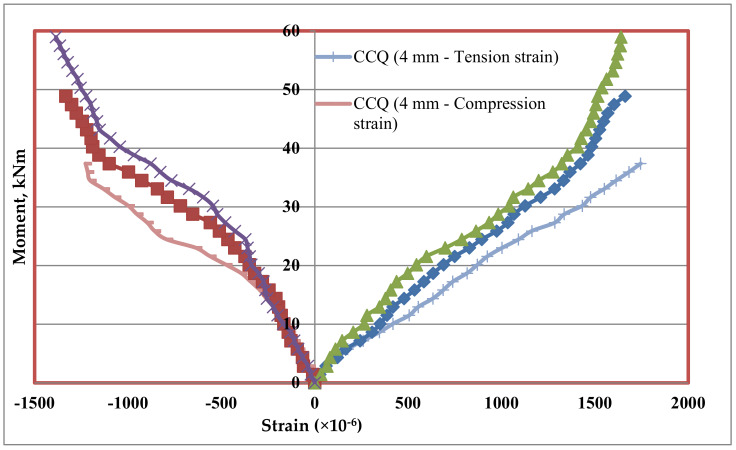
Strain distribution of SCS beams used with CCQ mix.

**Figure 21 materials-13-02444-f021:**
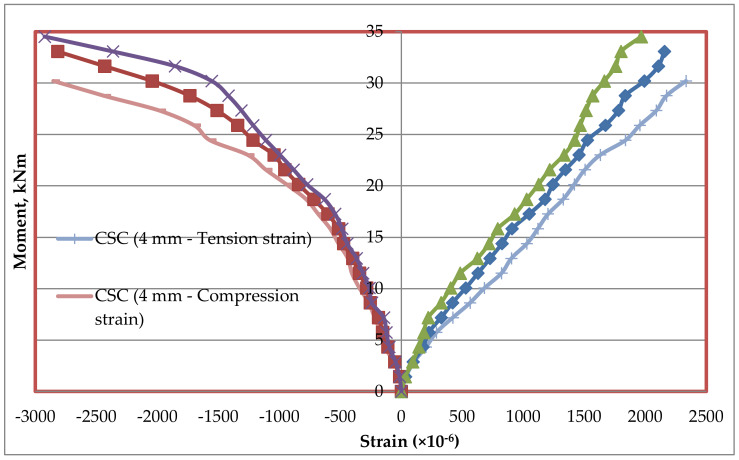
Strain distribution of SCS beams used with CSC mix.

**Figure 22 materials-13-02444-f022:**
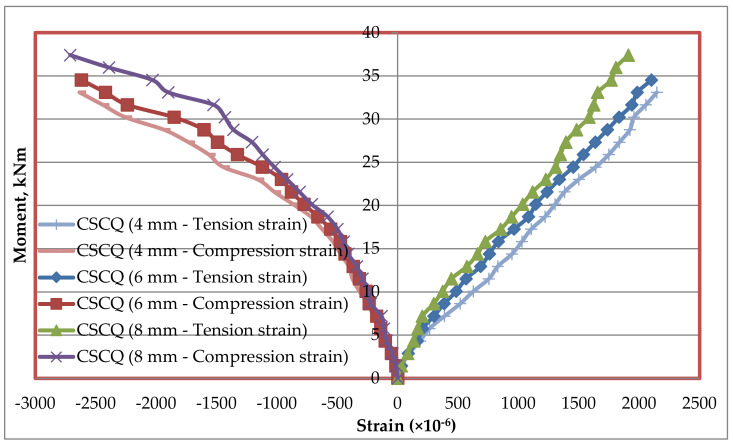
Strain distribution of SCS beams used with CSCQ mix.

**Figure 23 materials-13-02444-f023:**
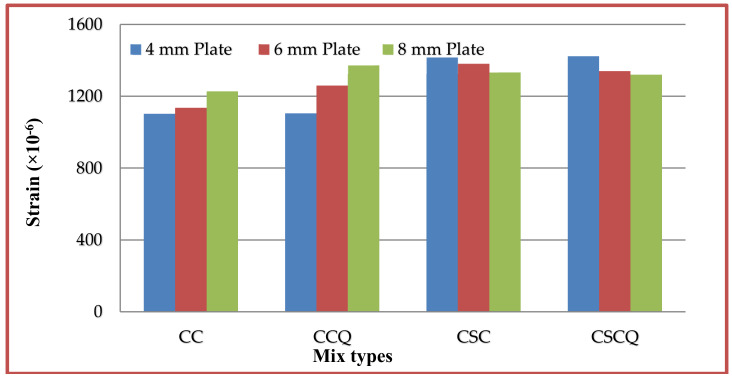
Bottom tension plates strains of different mixes.

**Figure 24 materials-13-02444-f024:**
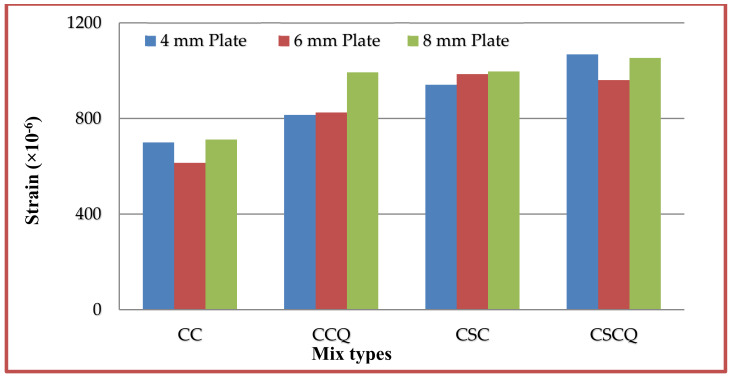
Top compression plates strains of different mixes.

**Table 1 materials-13-02444-t001:** Basic properties of fine and coarse aggregates used.

Physical and Mechanical Properties	CGS	CS	QD	RS
Maximum size (mm)	12.5	12.5	4.75	4.75
Water absorption (%)	–	24 (1.25)	–	–
Specific gravity	2.82 (0.03)	1.05–1.20 (0.06)	2.64 (0.02)	2.56 (0.03)
Fineness modulus	6.94 (0.03)	6.26 (0.04)	2.54 (0.01)	2.57 (0.03)
Bulk density (kg/m^3^)	1650 (3.09)	650 (4.08)	1700 (8.16)	1685 (4.08)
Shell thickness (mm)	12.5	2–8	–	–

Values in parentheses are standard deviation (SD) values of the results.

**Table 2 materials-13-02444-t002:** Mix proportions used for different concrete mixes.

Mix ID	Mix Proportion Cement: Fine Aggregate: Coarse Aggregate: Water	Cement Content
CC	1:2.22:3.66:0.55	320 kg/m^3^
CCQ	1:2.40:3.66:0.55	
CSC	1:1.47:0.65:0.42	510 kg/m^3^
CSCQ	1:1.58:0.62:0.42	

**Table 3 materials-13-02444-t003:** Fresh and hardened properties of different concrete mixes used.

Parameters	CC	CCQ	CSC	CSCQ
Target compressive strength (N/mm^2^)	25	25	25	25
Slump (mm)	9 (1.414)	0 (0)	6 (1.247)	0 (0)
Compaction factor	0.92 (0.017)	0.91 (0.025)	0.89 (0.021)	0.90 (0.026)
Fresh concrete density (kg/m^3^)	2495 (4.110)	2645 (5.249)	2100 (4.497)	2215 (4.546)
28-day hardened density (kg/m^3^)	2475 (5.735)	2600 (6.549)	1980 (3.300)	2150 (6.128)
28-day cube compressive strength (N/mm^2^)	30.18 (0.498)	32.78 (1.186)	26.83 (0.819)	29.05 (0.797)
28-day cylinder compressive strength (N/mm^2)^	24.15 (0.661)	26.22 (0.856)	21.46 (1.025)	23.24 (0.529)
Modulus of elasticity (N/mm^2^)	26,926 (12.329)	27,545 (26.158)	8780 (33.506)	9085 (23.343)

Values in parentheses are standard deviation (SD) values of the results.

**Table 4 materials-13-02444-t004:** Comparison of experimental moments with predicted results.

Mix ID	Plate Thickness (t) in (mm)	Depth of Neutral Axis (z) in (mm)	Predicted Moment (kNm)		Experimental M_exp_ (kNm)	Capacity Ratio (M_exp/_M_pl_)
			Elastic Theory	Plastic Theory		
			M_el_ (kNm)	M_pl_ (kNm)		
CC	4	70.81	31.88	33.90	34.52	1.02
CC	6	78.27	33.53	35.56	40.27	1.13
CC	8	83.24	33.94	35.87	48.90	1.36
CCQ	4	70.39	33.14	35.24	37.39	1.06
CCQ	6	77.86	33.52	35.56	48.90	1.38
CCQ	8	82.86	33.92	35.87	58.96	1.64
CSC	4	89.89	13.92	14.57	30.20	2.07
CSC	6	95.21	14.03	14.70	33.08	2.25
CSC	8	98.36	14.15	14.82	34.52	2.33
CSCQ	4	89.39	14.72	15.42	33.08	2.15
CSCQ	6	94.80	14.97	15.56	34.52	2.22
CSCQ	8	98.01	15.18	15.69	37.39	2.38

**Table 5 materials-13-02444-t005:** Comparison of experimental and predicted deflection of beams.

Mix ID	Plate Thickness (t) in (mm)	Predicted Deflection (mm) Δ_theo_	Experimental Deflection (mm) Δ_exp_	Δ_exp_/Δ_theo_
CC	4	5.84	5.40	0.92
CC	6	4.70	5.81	1.24
CC	8	4.36	6.29	1.44
CCQ	4	6.29	5.50	0.87
CCQ	6	5.68	5.72	1.01
CCQ	8	5.23	5.96	1.14
CSC	4	6.01	5.56	0.93
CSC	6	3.77	6.60	1.75
CSC	8	3.66	7.20	1.97
CSCQ	4	6.53	5.18	0.79
CSCQ	6	4.67	6.22	1.33
CSCQ	8	3.83	6.60	1.72

**Table 6 materials-13-02444-t006:** Ductility ratio of tested beams.

Mix ID	Plate Thickness (t) in (mm)	Experimental Yield Deflection (mm) Δ_y_	Experimental Ultimate Deflection (mm) Δ_u_	Ductility (µ) =Δ_u_/Δ_y_
CC	4	5.40	24.48	4.53
CC	6	5.81	26.14	4.50
CC	8	6.29	28.86	4.59
CCQ	4	5.50	22.54	4.10
CCQ	6	5.72	24.54	4.29
CCQ	8	5.96	26.50	4.45
CSC	4	5.56	27.40	4.93
CSC	6	6.60	28.70	4.35
CSC	8	7.20	30.10	4.18
CSCQ	4	5.18	26.14	5.05
CSCQ	6	6.22	27.30	4.39
CSCQ	8	6.60	29.18	4.42

**Table 7 materials-13-02444-t007:** Result of cracks formed details of the tested beams.

Mix ID	Plate Thickness (t) in (mm)	Experimental Crack Width at Yielding (mm)	Experimental Crack Width at Ultimate (mm)	Average Crack Spacing (mm)	Number of Cracks between the Load Point
CC	4	0.16	4.62	230	6
CC	6	0.14	3.81	235	6
CC	8	0.10	2.94	218	6
CCQ	4	0.15	4.40	206	6
CCQ	6	0.13	3.62	157	7
CCQ	8	0.08	2.84	137	7
CSC	4	0.22	4.86	183	7
CSC	6	0.18	4.29	198	8
CSC	8	0.14	4.08	149	8
CSCQ	4	0.20	4.61	143	8
CSCQ	6	0.16	3.99	147	7
CSCQ	8	0.13	3.83	173	7

**Table 8 materials-13-02444-t008:** Result of tension and compression steel plates’ strain of tested beams.

Mix ID	Plate Thickness (t) in (mm)	Strain on Bottom Tension Plate (×10^−6^)	Strain on Top Compression Plate (×10^−6^)
CC	4	1102	700
CC	6	1136	614
CC	8	1227	712
CCQ	4	1115	815
CCQ	6	1259	825
CCQ	8	1372	993
CSC	4	1416	941
CSC	6	1382	986
CSC	8	1333	996
CSCQ	4	1422	1068
CSCQ	6	1340	961
CSCQ	8	1320	1053

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
