# Peer review of "Behavior of Steel–Coconut Shell Concrete–Steel Composite Beam without and with Shear Studs under Flexural Load"

_materials, 2020, doi:10.3390/ma13112444_

Round 1

Reviewer 1 Report

The authors studied the double skin steel plate coconut shell concrete composite beam under flexure without and with shear studs. Few beam tests were performaed to study the flexural behaviour of the SCS sandwich beams with the new concrete materials.
These studies may provide information on the structural engineers working in this field. However, the manuscript can not be accepted in its current form, some major revisions are still required as the following;

(1) The introduction section needs to be rewritten. Some important and updated information in the filed of SCS sandwich structures as well as the SCS sandwich structures with different concrete materials are missing.
Below references may provide the updated information in this field.

Steel-UHPC-steel sandwich composite beams with novel enhanced C-channel connectors: Tests and analysis. Journal of Constructional Steel Research 2020; 170: 106077.
Flexural behaviours of steel-UHPC-steel sandwich beams with J-hook connectors. Journal of Constructional Steel Research 2020; 169: 106014.

(2) The research significance must be well illustrated in the introduciton, i.e., reasons of using the coconut shell concrete in the SCS sandwich structures.
(3) The shear studs used in Fig. 3 does not have the heads, pls offer details on the geometry of headed studs used.
(4) Some of the equations were just directly given without references, pls provide, e.g., Eqn. (1)~(4). If they were developed by the authors, pls offer more details
(5) The quality of the figures require further improvements, e.g., Fig. 9, Fig. 19-22.
(6) the language requires further improvements.
(7)To many self citations, pls delete some unnecessary self citations.

Reviewer 2 Report

1) Title: consider modifying the title, simplifying it. Use expressions more common to the scientific community (flexural loads instead of flexure, for instance)

2) typo of the word behaviour – line 22.

3) The English style of the text could be modified. In general, smaller and more direct sentences will facilitate the readers’ interpretation of the paper.

4) Introduction: Consider rearranging the introduction and enriching it. In the reviewer’ s opinion, one of the main topics of the paper is the use of coconut shell as an alternative raw material. However, few attention was given to this topic in the introduction. On the reviewer’s eyes, not only a description about how the CSC behaves should be given, but also the reasons why this material behaves this way.

    1. In their first paragraph, the authors inform that “there is a new form of sandwich construction called steel-concrete-steel developed in the recent years”. However, the concept of sandwich construction is well established in the scientific community since long times. The double skin composite SCS, specifically, dates back 1987 with the work of Narayanan et. al., which, btw, is not referenced by the authors.
    1. Check English consistency of lines 52-54.

5) Materials and properties:

    1. Consider modifying the tables’ labels to less generic expressions.
    2. Add standard deviation for the experimental results.
    3. Please, explain the reason why the modulus of elasticity of CSC and CSCQ is considerably lower than their counter-parts CC and CCQ.

6) Mechanics of beam deflection: Please, consider merging the sections 3, 4, 5 and 6 into a single section and better linking them. The way they are currently presented seems more like a collage of information than a continuous text (in the reviewer’s opinion).

7) Experimental investigation:

    1. Please, reconsider the necessity of all the images into this section.

8) Discussion on test results:

    1. Traditionally, discussion of results will only focus on analyzing the results attained during tests and comparing them with literature (when available). Please, consider re-positioning Figure 11 and its comments in the introduction section.
    2. Please explain why only CSC capacity ratios were above 2 (Table 4). Might it be directly related to the relative low modulus of elasticity for this material (presented on table 3)?
    3. Did the authors performed the experiments in a single beam of each type? If more than one experiment was performed for each type of beam, please add how the data is dispersed (standard deviation is a usual feature).

The reviewer will stop the review at this point and hopes that the comments might help the authors improving the quality of the paper. The topic is relevant to the scientific community, however the paper needs to be more mature and better described in order to be accepted for publication.

Round 2

Reviewer 1 Report

The authors have addressed the issues raised by the reviewer.

The careful proof reading is still required to improve the language of the manuscript.

Author Response

Point 1:

The careful proof reading is still required to improve the language of the manuscript.

Response 1:

As suggested by the reviewer, once again we requested another distinct native English speaking colleague to check the language. He suggested and corrected on careful proofreading and those corrections are highlighted using yellow colour in the revised manuscript (Revision 2). This is for the kind information of the Reviewer please.

Reviewer 2 Report

The reviewer recognizes the effort of authors to improve the paper's quality. New images were added, re-positioning of text was made and the research line was more clearly presented. However, some modifications are still necessary.

1 - The introduction must be improved. The new paragraphs added to it are not clear enough. There are redundant information, non-consistency in the citation style and the English style is not as expected.

2 - Typo on line 117 (dust instead of duct).

3 - Typo Figure 4 – please correct the word “Strain gauge"

4 - Please, elaborate why the difference of ratio capacity might be related to the lower modulus of elasticity of CSC and CSCQ. Just mentioning that it might be related is not clear enough. Even if this is an assumption, explain to the reader what has made you think this might be possible.

5 - Please, check that the comment 5a of reviewer 2 (first review) was stating: “Consider modifying the tables’ labels to less generic expressions”. The comment is still valid.

6 - Check the reference to figure 4b (line 293).

7 - Please check the English style throughout the text. 

Author Response

Point 1:

The introduction must be improved. The new paragraphs added to it are not clear enough. There are redundant information, non-consistency in the citation style and the English style is not as expected.

Response 1:

It is the duty of the authors to respond to the points raised by the reviewer. Though it is not an ethics to say about one reviewer comment to another reviewer, here being an author we must share one point, because this reviewer pointed out the paragraph added (because of another reviewer suggestion) in the earlier revision. For the kind information of the reviewer that the new paragraphs added in the revision 1 was introduced specially to respond the other reviewer suggestion, especially in that the other reviewer specifically sent to us 10 list of Journal papers (those 10 papers included in the reference list Ref.6 to Ref.15) to include in the introduction part and asked us to add in the reference list. Therefore, we the authors give respect to the reviewer comments we included. We also found that those works are somewhat advanced level research and extracting the some details from those papers may not be fit 100% in the introduction part of this manuscript. We tried our level best to extract some relevant points from those papers and included in that paragraph to the extent possible as an introduction part. However, being the authors we have to give respect to the comments posted by the reviewers in general. Though the perception of an individual will defer from person to person, we should give respect to their thought in this regard. We hope that the reviewer may also agree with us.

            However, further to respond the reviewer comment, some mores sentences are added in section 1.1 of the revision 2.  (Line 110 to 115 in the revision 2)

Point 2:

Typo on line 117 (dust instead of duct).

Response 2:

Corrected. Please refer the line 119 now in the revision (included track changes).

Thanks the reviewer for pointed out this typo error.

Point 3:

Typo Figure 4 – please correct the word “Strain gauge".

Response 3:

Corrected now in the revision (included track changes).

Once again thanks the reviewer for pointed out this typo error.

Point 4:

Please, elaborate why the difference of ratio capacity might be related to the lower modulus of elasticity of CSC and CSCQ. Just mentioning that it might be related is not clear enough. Even if this is an assumption, explain to the reader what has made you think this might be possible.

Response 4:

As asked by the Reviewer, to the extent possible we added some more points in this regard (line 538 to 544 in the revision 2). For the information reviewer, we authors not thought about in deep this, hence we thank the reviewer to give us some new thought to form a new research track towards this aspect.

Point 5:

Please, check that the comment 5a of reviewer 2 (first review) was stating: “Consider modifying the tables’ labels to less generic expressions”. The comment is still valid.

Response 5:

We were little bit confused about this particular comment since the reviewer was not satisfied the responses given by us in the revision 1. We discussed about this comment among ourselves and made changes. Still the reviewer asking us to check and hence we have changed the table labels in the revision 2 in another way to best of authors level. We hope that the reviewer may be satisfied with this.

Point 6:

Check the reference to figure 4b (line 293).

Response 6:

Corrected now in the revision (included track changes).

Once again thanks the reviewer for pointed out this correction.

Point 7:

Please check the English style throughout the text. 

Response 7:

 As suggested by the reviewer, once again we requested another distinct native English speaking colleague to check the language. He suggested and corrected on careful proofreading and those corrections are highlighted using yellow colour in the revised manuscript (Revision 2). This is for the kind information of the Reviewer please.
